# Nurse-Led Counseling Intervention of Postoperative Home-Based Exercise Training Improves Shoulder Pain, Shoulder Disability, and Quality of Life in Newly Diagnosed Head and Neck Cancer Patients

**DOI:** 10.3390/jcm11144032

**Published:** 2022-07-12

**Authors:** Yu-Long Hong, Tsung-Cheng Hsieh, Peir-Rong Chen, Shu-Chuan Chang

**Affiliations:** 1Institute of Medical Sciences, Tzu Chi University, Hualien 97004, Taiwan; ss171@ems.tcust.edu.tw (Y.-L.H.); tchsieh@mail.tcu.edu.tw (T.-C.H.); 2Department of Nursing, Tzu Chi University of Science and Technology, Hualien 97005, Taiwan; 3Department of Medicine, College of Medicine, Tzu Chi University, Hualien 97004, Taiwan; perong.chen@gmail.com; 4Department of Otolaryngology, Buddhist Tzu Chi Hospital, Hualien 97002, Taiwan; 5Department of Nursing, College of Medicine, Tzu Chi University, Hualien 97002, Taiwan; 6Nursing Committee, Buddhist Tzu Chi Medical Foundation, Hualien 97002, Taiwan

**Keywords:** head and neck cancer, exercise, shoulder pain, shoulder disability, quality of life

## Abstract

This randomized controlled trial investigated the effectiveness of the nurse-led counseling intervention (NLCI) of postoperative home-based exercise training (HBET) on functional outcomes in patients with newly diagnosed head and neck cancer (NDHNC). Forty NDHNC patients were randomly and equally divided into the control and intervention groups. Both groups received routine care, and were instructed to undergo a HBET program with 40 min moderate-intensity exercise 3–4 times per day for 12 weeks after their surgery. Only the intervention group received the NLCI with a bedside demonstration, coaching, consultation, and a weekly telephone follow-up. Shoulder pain (SP), shoulder disability (SD), and quality of life (QOL) scores were assessed using questionnaires at 2 weeks presurgery and at several timepoints postsurgery. Over the 12-week study period, all three scores remained relatively stable in the control group. By contrast, the SP, SD, and QOL scores significantly improved in the intervention group. The generalized estimating equation analysis revealed a significant time effect, group effect, and group–time interaction. The analysis of covariance revealed that all three scores significantly improved in the intervention group compared with those in the control group at 12 weeks postsurgery. We concluded that the NLCI of postoperative HBET improved the SP, SD, and QOL of NDHNC patients.

## 1. Introduction

Head and neck cancers (HNCs) are malignancies affecting the oral cavity, mucosal lip, pharynx, hypopharynx, nasopharynx, larynx, or salivary glands [1]. In 2018, HNCs were the seventh most common cancer worldwide, with 890,000 new cases and 450,000 deaths [2]. The prevalence of HNCs varies across different geological areas, and some regions, such as Taiwan, have high incidences [3]. HNCs are primarily treated with surgery alone or in combination with radiation therapy [4]. Patients with an advanced stage of cancer in the oral cavity may require chemoradiotherapy as the initial treatment of choice, but the therapy predisposes them to various complications [5]. Radical neck dissection and radiotherapy inevitably damage the nerves and muscles, resulting in various complications, such as shoulder pain (SP) and shoulder disability (SD), both of which may negatively impact the patients’ health-related quality of life (QOL) [4,6,7,8,9,10]. HNC patients undergoing head and neck surgery experience several postoperative complications, including impairments of speech and swallowing, and, consequently, are vulnerable to weight loss and reduced QOL [11]. Recently, robotic-assisted neck dissection has been suggested to be a less invasive technique for HNC patients, but it yielded similar functional and early oncologic outcomes compared to conventional neck dissection [12]. However, more clinical trials are required to establish this technique as an alternative standard [12]. Postoperative exercise training has been advocated to improve severe pain, disability, and QOL associated with HNC therapy [13,14,15]. Although some studies [16,17,18,19,20,21,22,23] reported that postoperative exercise training reduces SP and SD, and improves the functional capacity and QOL of HNC survivors, others [24,25,26] failed to detect its beneficial effects. As such, investigations optimizing the exercise program to improve these postoperative complications in patients with HNC are warranted.

Although the effectiveness of hospital-based exercise training has received interest in this research area [13,14], only a few studies have explored the benefits of home-based exercise training (HBET). HBET may have several advantages over hospital-based exercise programs, such as being relatively easy to perform, having less medical costs, being independent of instruments, saving transportation time to the hospital, and decreasing concern about hospital availability. However, patients’ health status, motivation, willingness, suitability for exercises, and adherence are some of the factors that may influence the effectiveness of exercise training in patients with HNC [15,27,28,29]. HNC survivors have many physical, Ning, emotional, and social issues that require consideration in the context of postoperative exercise training [15,27,29]. To this end, nurses are involved in the various stages of patient care and possess the necessary skills and knowledge to provide information, support, and coaching to patients with HNC [30]. Several aspects of nurse-led interventions aiming to improve postoperative outcomes in patients with HNC have been reported [30,31,32,33]. Additionally, some promising results regarding the beneficial effects of HBET on SP, SD, QOL, or functional capacity have been obtained in this patient population [21,22,23]. However, the benefits of the nurse-led counseling intervention of HBET on postoperative complications for patients with HNC remain to be explored.

In this study, we conducted a randomized controlled trial to investigate the effectiveness of the nurse-led counseling intervention of HBET in improving SP, SD, and QOL in patients with newly diagnosed HNCs. Patients in both study groups received routine care and were educated to perform postoperative HBET 2 weeks after surgery, subject to the approval of the primary surgeons. Only the intervention group received the nurse-led counseling intervention of HBET with a bedside demonstration, coaching, counseling, and a weekly telephone follow-up. The SP, SD, and QOL scores were measured before surgery and at several timepoints after surgery for 12 weeks for group comparisons.

## 2. Materials and Methods

### 2.1. Ethical Consideration and Setting

This randomized clinical trial was conducted at a medical center in eastern Taiwan. The study was conducted according to the guidelines of the Declaration of Helsinki and approved by the Research Ethics Committee of Hualien Tzu Chi Hospital and Buddhist Tzu Chi Medical Foundation (approval number IRB105-05-A). Informed consent was obtained from all subjects involved in the study. This study was registered on ClinicalTrials.gov with the identifier (NCT 05269342). The Consolidated Standards of Reporting Trials statement for the study design and reporting was adopted.

### 2.2. Participants and Study Design

This study was conducted from 1 August 2017 to 31 July 2018. Owing to the nature of the intervention, the participants or nurses who carried out the intervention were not blinded to the study. The total sample size was estimated to be 30 using G*Power with an effect size of 0.25, a two-tailed alpha of 0.05, and a power of 80%. With a 15% attrition rate, at least 20 participants in each group would be required for a total sample size of 40.

Inclusion criteria were as follows: (1) aged >20 years old, (2) newly diagnosed with HNC by a physician, (3) scheduled to receive surgery, (4) had no serious complications, (5) had no history of mental illness, (6) had no comorbidities, (7) had no sensory–cognitive problems, and (8) could communicate verbally and respond to the questionnaires. Exclusion criteria were as follows: (1) could not communicate verbally and respond to the questionnaires; (2) had a history of shoulder disorders before surgery, including shoulder pain, tendinitis, tendon rupture, and shoulder capsulitis; (3) had pre-existing shoulder pain, dysfunction, or weakness, including neuropathy or arthropathy during preoperative assessment; (4) had any pre-existing disorders that could influence physical activity; and (5) had concurrent chemoradiotherapy.

After undergoing otolaryngologist evaluation and providing informed consent, 40 participants were randomly assigned by dedicated nurses to the control (*n* = 20) or intervention group (*n* = 20) using a computer-generated code. The allocation was concealed to the participants who also received written assurance that the postcancer care provided by the HNC specialist would be the same in both groups. All participants did not receive chemoradiotherapy during the study period.

### 2.3. Routine Care

All patients received standard care, medical treatments, and continued healthcare, and were provided with health education, an educational pamphlet containing various aspects of postsurgery self-care information including the HBET manual/video, and an individual education session that explained the methods of the HBET before surgery.

### 2.4. HBET

The HBET manual and video were provided by the Taiwan Cancer Foundation (https://elearning.canceraway.org.tw/page.asp?IDno=1541, accessed on 1 May 2022). The HBET was developed and validated by cancer and exercise experts to encourage patients with HNC to perform exercises by themselves at home [34]. The HBET included: (1) active resistance exercise for the major muscles of the upper and lower limbs by stretching, (2) active and passive range of motion exercise of the shoulder joints by overhead shoulder flexion, and (3) open-mouth training, hip flexion, knee extension, and hip abduction. The participants were instructed to hold each training movement for 10 s 10 times. The total training duration of HBET lasted for approximately 40 min. The participants were instructed to initially perform HBET at least once daily and gradually increased to four times a day. This training was classified as a moderate-intensity exercise program that was tailored for patients with HNC and mainly focused on neck extension and shoulder girdle relaxation.

### 2.5. Nurse-Led Counseling Intervention

The nurse-led counseling intervention aimed to help patients manage themselves physically and psychologically by giving education, coaching, care, and support. Prior to the study, the dedicated nurses from the otolaryngology department who were responsible for the care of HNC patients underwent a 1-month specific training for HBET, which was supervised by specialists from the department of surgical oncology and rehabilitation. The dedicated nurses were also well educated so they could properly consider specific issues regarding HBET and ensure the safety of the exercises. After training, the dedicated nurses visited the intervention group in the ward to provide guidance and demonstrate each training movement in HBET before and 2 weeks after the surgery. All the participants were instructed to practice each training movement in HBET before the surgery, and with approval from the primary surgeon, perform the full scale of HBET 2 weeks after the surgery. The dedicated nurses provided a weekly telephone follow-up during the study period to track the participants’ exercise performance and provide support and consultation. All participants were instructed to record their daily exercise activities at home. The participants who performed HBET at a reduced frequency were reminded of the necessary precautions and monitored for any complications. All participants were monitored daily for any complications during the course of HBET.

### 2.6. Measures

Demographic and baseline clinical characteristics, including age, sex, educational level, and cancer stage, were recorded. Primary outcome data were collected 2 weeks before surgery (baseline) and then 2, 4, 8, and 12 weeks after surgery. SP and SD were assessed using a questionnaire, which was developed by Roach et al. [35]. This questionnaire was divided into two assessment parts for SP (a total score of 50) and SD (a total score of 50) consisting of five and eight items, respectively. Each item was scored on a scale of 0 to 10, in which a high score indicated the severity of SP or SD. QOL was assessed by using the University of Washington Quality of Life Questionnaire (UW-QOL), one of the most commonly used questionnaires in HNC [36]. This questionnaire includes the assessments of pain, appearance, activities, entertainment, work, swallowing ability, chewing ability, talking ability, shoulder functions, taste, and saliva status, and each item has a Likert scale ranging from 0 (worst) to 100 (optimal). Questionnaire data were collected before surgery and after surgery during the weekly telephone follow-up and outpatient visit evaluation.

### 2.7. Statistical Analysis

The Kolmogorov–Smirnov test was used to check the distribution of continuous variables, which were compared with the Student’s t-test and presented as mean ± standard deviation (SD). Pearson’s Chi-square test or Fisher’s exact test were used to compare the categorical variables, which were presented as frequency and percentages. For SP, SD, and QOL scores, a generalized estimating equation (GEE) model, including the covariates of the group, time, group–time interaction, and baseline, was constructed to evaluate the effects of group and group–time interactions. If an interaction was observed, then an analysis of the covariance (ANCOVA) model, including the covariates of the group and baseline, was conducted at each time point to evaluate the group effects. Statistical data were analyzed using the SPSS statistics version 20.0. P values of less than 0.05 were considered statistically significant for all analyses.

## 3. Results

### 3.1. Baseline Patient Characteristics

This randomized controlled trial enrolled 40 patients with HNC who were equally divided into the control and intervention groups. Figure 1 shows the flowchart of patient deposition. All the enrolled patients completed the experimental protocol during the 12-week study period. No adverse events were reported, and all the participants provided complete data. Table 1 shows the comparisons of baseline demographic data between these two study groups. No significant differences in all measured parameters were found between these two study groups.

### 3.2. Effects of the Nurse-Led Counseling Intervention of HBET: GEE Analyses

Figure 2 shows the changes in the absolute mean SP, SD, and UW-QOL scores in the two study groups over time. Between-group comparisons revealed a significant difference in the SD (*p* < 0.001) or UW-QOL (*p* < 0.001) score measured at 2 weeks before surgery (baseline), but not in the SP score. After surgery, the SP (Figure 2A) and SD (Figure 2B) scores showed a gradually decreasing trend, and the QOL score (Figure 2C) showed a gradually increasing trend over the 12-week study period in the intervention group. However, these scores remained relatively stable in the control group over the study period (Figure 2). Table 2 shows the results of the GEE analyses of the three scores in relation to the group and time. Significant group effect, time effect, and group–time interactions were observed at 12 weeks postsurgery. The group effects of SP, SD, and UW-QOL scores were β = 22.06 (*p* < 0.01), β = −2.30 (*p* < 0.01), and β = 422.9 (*p* < 0.01), respectively. The time effects of SP, SD, and UW-QOL scores were β = 2.19 (*p* < 0.01), β = −0.21 (*p* < 0.01), and β = 76.8 (*p* < 0.01), respectively. The group–time effects of SP, SD, and UW-QOL scores were β = 6.94 (*p* < 0.01), β = 0.07 (*p* < 0.05), and β = −105.2 (*p* < 0.01), respectively.

### 3.3. Effects of the Nurse-Led Counseling Intervention of HBET: ANCOVA Analyses

ANCOVA analyses were further performed using the 12-week outcome measures (three scores) as the dependent variable and the groups as the fixed effect and controlling for presurgery scores. The ANCOVA results in Table 3 revealed significant differences in all three adjusted mean scores measured at 12 weeks postsurgery between the two study groups (SP score: β = −8.09, *p* < 0.01; SD score: β = −19.49, *p* < 0.01; UW-QOL score: β = −143.48, *p* < 0.01). The changes in the absolute values of the adjusted mean scores of the two study groups over time are shown in Figure 3. Improvements in the adjusted mean SD score became apparent 4 weeks postsurgery, and improvements in the SP and QOL scores became apparent 8 weeks postsurgery in the intervention group compared with those in the control group.

## 4. Discussion

In this study, we revealed that HBET with a nurse-led counseling intervention for 12 weeks was effective in improving SP, SD, and QOL in patients newly diagnosed with HNC as evidenced by the results from the GEE and ANCOVA analyses. Conversely, the effectiveness of the HBET was not apparent in participants without the nurse-led counseling intervention. The demographic characteristics of these two study groups were similar, and no dropout was recorded during the intervention and evaluation. Our results, thus, substantiated the hypothesis that the HBET program could be performed by patients themselves at home to improve these complications, providing these patients received adequate training, consultation, care, and support from the nurses.

Several guidelines endorse the implementation of exercise programs for patients with cancer to promote high-quality cancer care [37]. The mechanism behind this beneficial effect remains largely unclear. Previous studies [38] have shown that voluntary exercise results in an influx of immune cells in tumors and significant reductions in tumor incidence and growth in mouse models. These investigators [38] proposed that the link between exercise and the immune system can be exploited in cancer therapy and that exercise may not just be “healthy”, but may in fact be therapeutic. Previous studies on the effectiveness of postoperative exercise training on SP, SD, QOL, and functional capacity in HNC survivors revealed positive results [13,14,16,17,18,19,20,21,22,23,24,25,26]. Although many authors have studied the effectiveness of hospital-based exercise training in this research area [16,17,18,19,20,24,25,26], relatively few investigations have been conducted to explore the clinical benefit of HBET [21,22,23]. HBET has certain advantages, such as its simplicity to perform, low medical cost, and reduction in transportation time to the hospital. One additional advantage is that in areas with limited medical resources, HBET could meet the needs of HNC survivors and provide them with an alternative choice of exercise training. The HBET used in the present study was a moderate-intensity exercise program tailored for patients with HNC and is recommended by the Taiwan Cancer Foundation to encourage patients with HNC to exercise by themselves at home. Samuel et al. [21] reported that patients with HNC undergoing chemoradiotherapy who did not receive any exercise training exhibited a decline in functional capacity and QOL, whereas those who received a structured HBET program showed improvements in both indices after a 6-week study period. Su et al. [22] found that HBET with a telephone follow-up for 12 weeks was not inferior to outpatient physical therapy regarding improvements in SP, SD, and functional capacity in patients with HNC. Do et al. [23] showed that although HBET was effective in improving QOL, shoulder function, and pain relief, a hospital-based exercise program by a supervised physical therapist had superior effects on the physical function of the neck and shoulder in HNC survivors. Although the designs and HBET programs of these studies were dissimilar to those in the present study, all these and our findings provide clinical evidence that HBET is feasible to implement and is effective in improving SP, SD, and QOL.

In exercise training programs, HNC survivors are considered a special population, because HNC is one of the most psychologically and emotionally traumatic cancer diseases [15,29]. HNC survivors have many physical, psychological, emotional, and social issues that may impede postoperative exercise training [15,27,29]. For the HBET program, patients’ health status, motivation, willingness, and suitability for exercises are some of the major issues to be considered, because patients need to persistently perform the training on schedule at home by themselves [27,28]. To this end, nurses possess the necessary skills and knowledge to provide information, support, and coaching to patients with HNC [30]. Nurses are medical professionals who have frequent in-hospital interactions with patients and can thereby establish a relationship to earn the trust of patients. Nurse-led interventions in patients with HNC have been shown to improve postoperative patient outcomes, including psychosocial adjustment, health-related QOL, nutrition impact symptoms, and postsurgical complications [30,31,32,33]. The findings of this study highlight the importance of nurse-led coaching, care, and support in promoting the effectiveness of HBET in patients with HNC.

The randomized controlled trial design was the strength of this study. However, it also had some limitations that need to be addressed. First, this study was a single-center investigation and involved a relatively small sample size. Thus, future multicenter investigations with large sample sizes are warranted to confirm our findings. Second, the adherence rate was not monitored because it was not a part of the study objective. Data on the adherence rate can be readily recorded for hospital-based exercise programs, but may not be acknowledged as accreditable data for HBET programs. Collecting data on or an attempt to increase the adherence rate sometimes may put stress on patients, leading to program dropout. In this work, we intentionally provided care and encouragement only to our patients during the telephone follow-up. Third, this study included newly diagnosed patients with HNC and was conducted with a relatively short follow-up time with no concurrent chemoradiotherapy. The generalization of our findings to other patient populations or to a long follow-up time requires further investigation.

## 5. Conclusions

Our results suggested that the nurse-led counseling intervention of postoperative HBET improved SP, SD, and QOL in patients with newly diagnosed HNC. The HBET program was safe and tolerable, and the coaching, consultation, care, and support from the nurses were crucial for the effectiveness of the HBET in these patients. This HBET with a nurse-led consultation and follow-up may be implemented in the postsurgery care of HNC survivors.

## Figures and Tables

**Figure 1 jcm-11-04032-f001:**
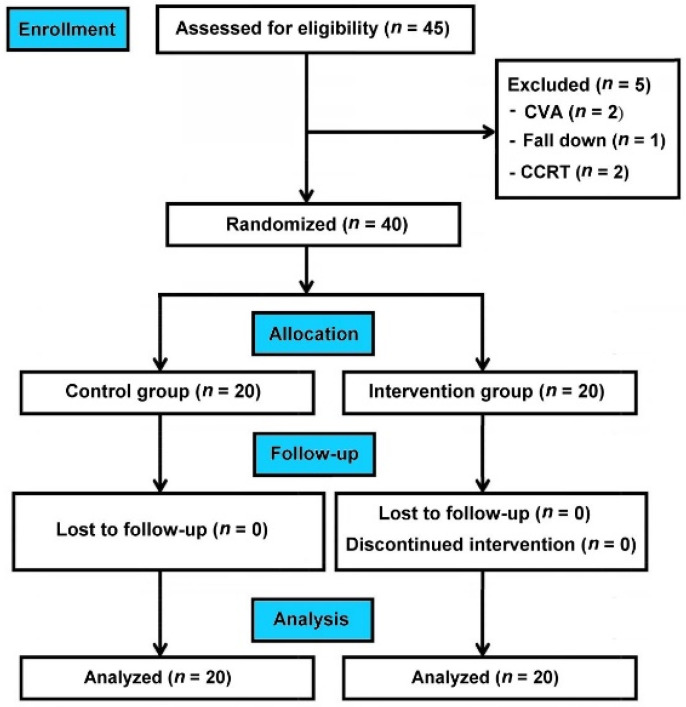
CONSORT flow diagram. CONSORT, Consolidated Standards of Reporting Trials; CVA, cerebrovascular accident; CCRT, concurrent chemoradiotherapy.

**Figure 2 jcm-11-04032-f002:**
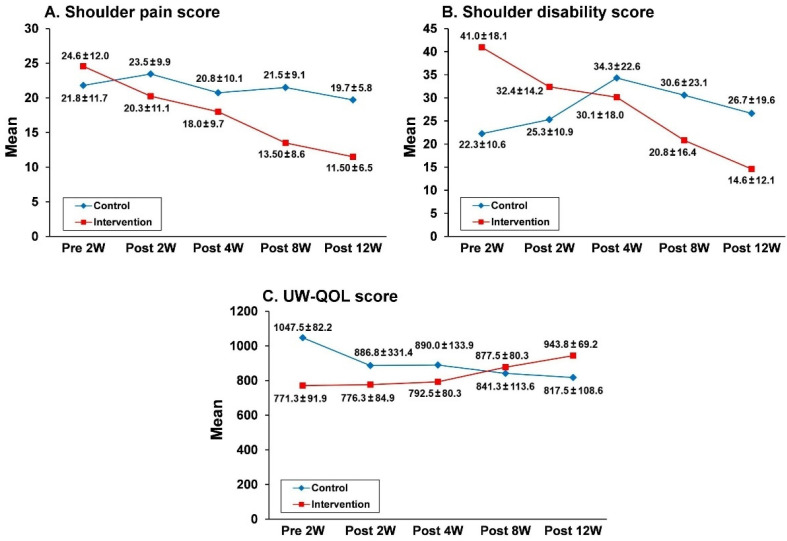
Changes in mean scores for shoulder pain (**A**), shoulder disability (**B**), and UW-QOL (**C**) over time during the study period. Mean scores were expressed as absolute values. UW-QOL, University of Washington Quality of Life Questionnaire. Post 2W, 4W, 8W, and 12W represent assessments at 2, 4, 8, and 12 weeks after surgery, respectively.

**Figure 3 jcm-11-04032-f003:**
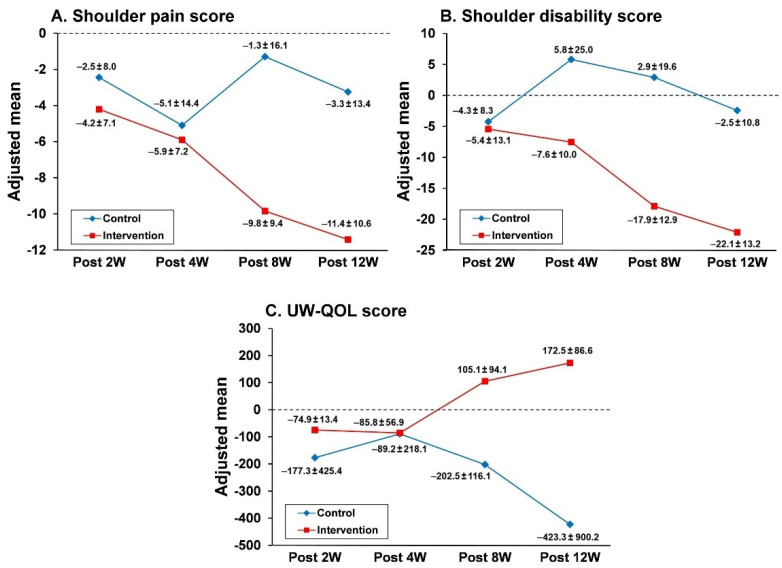
Changes in adjusted mean scores for shoulder pain (**A**), shoulder disability index (**B**), and UW-QOL (**C**) over time during the study period. Adjusted mean scores were estimated using the analysis of covariance (ANCOVA) model in Table 3 and were expressed as absolute values. UW-QOL, University of Washington Quality of Life Questionnaire. Post 2W, 4W, 8W, and 12W represent assessments at 2, 4, 8, and 12 weeks after surgery, respectively.

**Table 1 jcm-11-04032-t001:** Comparison of baseline demographic data and clinical characteristics between the control and intervention groups.

		Total	Control	Intervention	*p*-Value
Variables		(*n* = 40)	(*n* = 20)	(*n* = 20)	
Age (year)		58.2 ± 10.1	58.1 ± 10.6	58.4 ± 9.8	0.58
Length of stay (day)		11.1 ± 2.4	11.1 ± 2.4	11.2 ± 2.4	0.87
Sex	M	30 (75)	16 (80)	14 (70)	0.47
	F	10 (25)	4 (20)	6 (30)	
Cancer stage	I	26 (65)	15 (75)	11 (55)	0.40
	II	8 (20)	2 (10)	6 (30)	
	III	6 (15)	3 (15)	3 (15)	
Elective neck dissection		32 (80)	17 (85)	15 (75)	0.69
Therapeutic neck dissection		8 (20)	3 (15)	5 (25)	0.69
Nodal positivity		8 (20)	3 (15)	5 (25)	0.69
Age	<40 years	3 (7)	1 (5)	2 (10)	0.86
	41–50 years	6 (15)	5 (25)	1 (5)	
	51–60 years	14 (35)	4 (20)	10 (50)	
	>61 years	17 (43)	10 (50)	7 (35)	
Work	No	7 (17)	2 (10)	5 (25)	0.76
	Yes	33 (83)	18 (90)	15 (75)	
Education	<9 years	20 (50)	7 (35)	13 (65)	0.06
	>12 years	20 (50)	13 (65)	7 (35)	
Marriage	Single	12 (30)	5 (25)	7 (35)	0.50
	Married	28 (70)	15 (75)	13 (65)	
Smoking	Never	9 (23)	3 (15)	6 (30)	0.41
	Abstained	16 (40)	12 (60)	4 (20)	
	<1 pack/day	3 (7)	1 (5)	2 (10)	
	>1 pack/day	12 (30)	4 (20)	8 (40)	
Alcohol	Never	25	4 (20)	6 (30)	0.46
	Abstained	10 (25)	3 (15)	7 (35)	
	<250 mL/day	13 (33)	11 (55)	2 (10)	
	>250 mL/day	7 (17)	2 (10)	5 (25)	
Betel nut	Never	7 (17)	4 (20)	3 (15)	0.41
	Abstained	24 (60)	13 (65)	11 (55)	
	<10/day	2 (6)	0 (0)	2 (10)	
	>10/day	7 (17)	3 (15)	4 (20)	
Exercise activity	Yes	14 (35)	6 (30)	8 (40)	0.52
	No	26 (65)	14 (70)	12 (60)	
Chronic disease	No	23 (58)	10 (50)	13 (65)	0.35
	Yes	17 (42)	10 (50)	7 (35)	

Data are presented as mean ± SD or *n* (%). Education < 9 years included patients with a degree up to junior high school. Education > 12 years included patients with high school degree or above. Exercise activity was defined as twice a week for at least 30 min, with a heartbeat of at least 130 beats per minute as defined by the World Health Organization. Chronic diseases included patients with one or more of the four main types of noninfectious diseases as defined by the World Health Organization. These are (1) cancer, (2) cardiovascular disease, including cerebrovascular disease, heart failure, and coronary artery disease; (3) chronic respiratory diseases, such as asthma and chronic obstructive pulmonary disease; and (4) diabetes (type 1 diabetes, type 2 diabetes, prediabetes, and gestational diabetes).

**Table 2 jcm-11-04032-t002:** The GEE analyses of shoulder pain, shoulder disability, and UW-QOL scores in relation to group and time (*n* = 40).

	SP				SD				UW-QOL			
Parameters	β	SE	95% CI	*p*	β	SE	95% CI	*p*	β	SE	95% CI	*p*
Intercept	11.6	7.6	−3.2 to 26.4	<0.01 **	1.1	1.1	−1.0 to 3.2	<0.01 **	639.9	27.6	585.8 to 693.9	<0.01 **
Group	22.1	5.1	11.9 to −32.2	<0.01 **	−2.3	1.9	−6.1 to 1.5	<0.01 **	422.9	35.44	353.5 to 492.5	<0.01 **
Time (Post 12W)	2.2	1.3	−9.8 to 4.0	<0.01 **	−0.2	0.2	−0.6 to 0.2	<0.01 **	76.8	7.6	61.8 to 91.8	<0.01 *
Group × Time (Post 12W)	6.9	1.5	1.5 to −9.8	<0.01 **	0.1	0.4	−0.6 to 0.8	<0.05 *	-105.2	9.8	−124.4 to −85.9	<0.01 **

CI, confidence interval; GEE, generalized estimating equation; SP, shoulder pain; SD, shoulder disability; UW-QOL, University of Washington Quality of Life Questionnaire; Post 2W, 4W, 8W, and 12W represent assessments at 2, 4, 8, and 12 weeks after surgery, respectively. * *p* < 0.05, ** *p* < 0.01.

**Table 3 jcm-11-04032-t003:** ANCOVA analyses of changes in the adjusted mean scores for shoulder pain, shoulder disability, and UW-QOL at different timepoints after surgery (*n* = 40).

	Post 2W				Post 4W				Post 8W				Post 12W			
Variable	Adjusted Mean	β	SE	*p*	Adjusted Mean	β	SE	*p*	Adjusted Mean	β	SE	*p*	Adjusted Mean	β	SE	*p*
**SP** **score**																
Intervention	−4.21	−1.68	2.60	0.52	−5.98	−0.86	3.41	0.25	−9.85	−8.55	2.60	<0.01 **	−11.39	−8.09	1.87	<0.01 **
Control	−2.53				−5.11				−1.29				−3.30			
**SD score**																
Intervention	−5.36	−1.03	3.24	0.75	−7.60	−13.36	6.66	<0.05 *	−17.96	−20.91	6.11	<0.01 **	−22.07	−19.49	5.26	<0.01 **
Control	−4.33				5.75				2.96				−2.57			
**UW-QOL score**																
Intervention	−74.90	−102.44	178.8	0.32	−85.78	−3.43	119.89	0.97	105	−90.65	73.80	<0.01 **	172.5	−143.48	269.0	<0.01 **
Control	−177.34				−89.21				−202.5				−423.25			

ANCOVA, analysis of covariance; SE, standard; SP, shoulder pain; SD, shoulder disability; UW-QOL, University of Washington Quality of Life Questionnaire; Post 2W, 4W, 8W, and 12W represent assessments at 2, 4, 8, and 12 weeks after surgery, respectively. * *p* < 0.05, ** *p* < 0.01.

## Data Availability

The data are available on request from the corresponding author. The data of this study are not publicly available since the data consist of information that could compromise research participants’ privacy and consent.

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
