# Peer review of "Nurse-Led Counseling Intervention of Postoperative Home-Based Exercise Training Improves Shoulder Pain, Shoulder Disability, and Quality of Life in Newly Diagnosed Head and Neck Cancer Patients"

_jcm, 2022, doi:10.3390/jcm11144032_

Round 1
Reviewer 1 Report
- line 39, Cancer of the oral cavity for the very frequently advanced stages to diagnostics requires in the first instance a chemoradiotherapy treatment which represents the treatment of choice but predisposes to different complications., please discuss and cite doi:10.1016/j.anl.2021.05.007.
- line 42, Robotic surgery propose a less invasive tecnique tnah traditional surgery. However, the similar oncologic and functional outcomes compared with the open procedure require more prospective, controlled, multicenter studies to establish robotic-assisted neck dissection as an alternative standard and to justify its added costs beyond the cosmetic advantages. please discuss and cite doi:10.1016/j.ijsu.2015.11.022
- line 48, patients undergoing head and neck surgery have a greater risk of postoperative complications due to alterations in speech, and swallowing and consequently are often subject to weight loss and reduced quality of life., please discuss and cite doi:10.1007/s00405-016-4177-0.
- why in the methods the follow up was not 2022 and only 2018?
- HBET guidelines or reference should be cited
- the consort model diagram shoud be adopted.
- table 2 quality is poor
- line 325, Exercise improves functional capacity and patient-reported outcomes across a range of cancer diagnoses. The mechanisms behind this protection have been largely unknown, but exercise-mediated changes in body composition, sex hormone levels, systemic inflammation, and immune cell function have been suggested to play a role. Voluntary exercise leads to an influx of immune cells in tumors, and a more than 60% reduction in tumor incidence and growth across several mouse models. Given the common mechanisms of immune cell mobilization in mouse and man during exercise, with a link between exercise and the immune system can be exploited in cancer therapy in particular in combination with immunotherapy. Thus, exercise may not just be "healthy" but may in fact be therapeutic, please discuss and cite doi:10.1007/s00262-017-1985-z
Author Response
Responses to Comments from Reviewer 1 (jcm-1771808)
We would like to thank the reviewers for their extensive assessment of our manuscript, and for important and helpful comments and suggestions. We have responded to all the reviewer’s comments in a point-by-point fashion and have revised the manuscript accordingly. The revised portions are indicated by “Track Changes”. We hope that the changes made will be considered satisfactory.
Comment 1: line 39, Cancer of the oral cavity for the very frequently advanced stages to diagnostics requires in the first instance a chemoradiotherapy treatment which represents the treatment of choice but predisposes to different complications., please discuss and cite doi:10.1016/j.anl.2021.05.007.
Response: We thank the reviewer for providing this suggestion and helpful information. We have added a sentence in the introduction section (lines 39-41) to discuss this issue. The sentence reads as: “Patients with advanced stage cancer in the oral cavity may require chemoradiotherapy as the initial treatment of choice, but the therapy predisposes to various complications [6].” A new reference (Auris Nasus Larynx. 2022;49:117-125; reference 6) as suggested by the reviewer has been added to support this statement. The sequence of the citation has been amended accordingly.
Comment 2: line 42, Robotic surgery propose a less invasive technique than traditional surgery. However, the similar oncologic and functional outcomes compared with the open procedure require more prospective, controlled, multicenter studies to establish robotic-assisted neck dissection as an alternative standard and to justify its added costs beyond the cosmetic advantages. please discuss and cite doi:10.1016/j.ijsu.2015.11.022
Response: We thank the reviewer for providing this suggestion and helpful information. We have added sentences in the introduction section (lines 47-50) to discuss this issue. These sentences read as: “Recently, robotic-assisted neck dissection has been suggested to be a less invasive technique for HNC patients, but yield similar functional and early oncologic outcomes compared to conventional neck dissection [12]. However, more clinical trials are required to establish this technique as an alternative standard [12].” A new reference (Int J Surg. 2016;25:24-30; reference 12) as suggested by the reviewer has been added to support these statements. The sequence of the citation has been amended accordingly.
Comment 3: line 48, patients undergoing head and neck surgery have a greater risk of postoperative complications due to alterations in speech, and swallowing and consequently are often subject to weight loss and reduced quality of life., please discuss and cite doi:10.1007/s00405-016-4177-0.
Response: We thank the reviewer for providing this suggestion and helpful information. We have added a sentence in the introduction section (lines 44-47) to discuss this issue. The sentence reads as: “HNC patients undergoing head and neck surgery have several postoperative complications including impairments of speech and swallowing, and consequently are vulnerable to weight loss and reduced QOL [11].” A new reference (Eur Arch Otorhinolaryngol. 2016;273:4359-4368; reference 11) as suggested by the reviewer has been added to support this statement. The sequence of the citation has been amended accordingly.
Comment 4: why in the methods the follow up was not 2022 and only 2018?
Response: We thank the reviewer for allowing us to further explain this issue. This study was approved by the Institutional Review Board (IRB) with a follow up period of 12 weeks till July 31, 2018. Our study complied with the follow up period approved by IRB. Most of other studies [16, 17, 18, 19, 22, 28; references in the manuscript] investigating the effectiveness of exercise training on functional capacity and quality of life in HNC patients had the same follow up period (12 weeks). Some studies had 6 weeks [21], 4 weeks [23] or 8 weeks [26] of follow up.
Comment 5: HBET guidelines or reference should be cited.
Response: We thank the reviewer for this suggestion. We have cited a new reference for HBET (reference 34) used in this study and have indicated this citation in the text (line 125).
Comment 6: the consort model diagram should be adopted.
Response: We thank the reviewer for this suggestion. We have redrawn figure 1 to fit the consort model (new Figure 1).
Comment 7: table 2 quality is poor
Response: We thank the reviewer for this suggestion. We have replaced tables 2 and 3 by new tables with good quality (new Tables 2 and 3).
Comment 8: line 325, Exercise improves functional capacity and patient-reported outcomes across a range of cancer diagnoses. The mechanisms behind this protection have been largely unknown, but exercise-mediated changes in body composition, sex hormone levels, systemic inflammation, and immune cell function have been suggested to play a role. Voluntary exercise leads to an influx of immune cells in tumors, and a more than 60% reduction in tumor incidence and growth across several mouse models. Given the common mechanisms of immune cell mobilization in mouse and man during exercise, with a link between exercise and the immune system can be exploited in cancer therapy in particular in combination with immunotherapy. Thus, exercise may not just be "healthy" but may in fact be therapeutic, please discuss and cite doi:10.1007/s00262-017-1985-z
Response: We thank the reviewer for providing this excellent suggestion and helpful information. We have added a paragraph in the discussion section (lines 283-288) to discuss this issue. These sentences read as: “The mechanisms behind this beneficial effect remains largely unclear. Previous studies [38] have shown that voluntary exercise results in an influx of immune cells in tumors and significant reductions in tumor incidence and growth in mouse models. These investigators [38] proposed that the link between exercise and the immune system can be exploited in cancer therapy and that exercise may not just be "healthy" but may in fact be therapeutic.” A new reference (Cancer Immunol Immunother. 2017;66:667-671; reference 38) as suggested by the reviewer has been added to support these statements. The sequence of the citation has been amended accordingly.

Reviewer 2 Report
You have conducted an RCT to evaluate the effect of shoulder exercises after treatment of head and neck cancer patients, with positive results. Shoulder disorders occur in patients with head and neck cancer who have undergone neck dissection. Therefore, shoulder disorders are more likely to occur in patients with advanced cancer. The major complication in patients with advanced head and neck cancer is aspiration pneumonia. This article is highly novel and useful to the reader because it identifies one intervention for shoulder disorders that is easily overlooked by the medical profession. The inclusion of a quality-of-life assessment in addition to symptom assessment is also a good consideration of recent trends in clinical research.
The intervention is beneficial to patients, and there are no special remarks regarding the study design, statistical methods, or presentation of results.
Author Response
Responses to Comments from Reviewer 2 (jcm-1771808)
Comment 1: This article is highly novel and useful to the reader because it identifies one intervention for shoulder disorders that is easily overlooked by the medical profession. The inclusion of a quality-of-life assessment in addition to symptom assessment is also a good consideration of recent trends in clinical research. The intervention is beneficial to patients, and there are no special remarks regarding the study design, statistical methods, or presentation of results.
Response: We thank the reviewer for the time and effort in reviewing our manuscript and for the positive feedback.

Reviewer 3 Report
Hong et al. perform a randomized controlled trial on 40 patients undergoing surgery for head and neck cancer, for nurse-led counseling on home based exercises. Overall, I find this an interesting study with encouraging results showing impact on patient outcomes. The trial design and statistics seem appropriate. Language is overall appropriate, with minor grammatical and spelling errors. Some comments:
1. Can the authors comment on extend of neck dissection in these patients? Was level 2B routinely taken? Was the nodal positivity status, extent of neck dissection amongst the cohort and was this similar between the 2 groups?
2. Can the authors describe the nursing profile? Were these outpatient nurses within the otolaryngology department, nurses focused on head and neck cancer patienets?
3. Is there a reason error bars were not included in Figures 2 and 3?
4. Was ECOG/Zubrod score similar between the 2 groups? ECOG/Zubrod is usually noted in quality of life studies.
Author Response
Responses to Comments from Reviewer 3 (jcm-1771808)
We would like to thank the reviewers for their extensive assessment of our manuscript, and for important and helpful comments and suggestions. We have responded to all the reviewers’ comments in a point-by-point fashion and have revised the manuscript accordingly. The revised portions are indicated by “Track Changes”. We hope that the changes made will be considered satisfactory.
Comment 1: Overall, I find this an interesting study with encouraging results showing impact on patient outcomes. The trial design and statistics seem appropriate. Language is overall appropriate, with minor grammatical and spelling errors.
Response: We thank the reviewer for the time and effort in reviewing our manuscript and for the positive feedback. The grammatical and spelling errors have been carefully checked and amended throughout the text.
Comment 2: Can the authors comment on extend of neck dissection in these patients? Was level 2B routinely taken? Was the nodal positivity status, extent of neck dissection amongst the cohort and was this similar between the 2 groups?
Response: Yes. Almost all our patients underwent level II b dissection. In response to the reviewer’s suggestion, we have added these clinical characteristics in the Table 1. As shown, these clinical characteristics were not significantly different between the 2 group.
Comment 3: Can the authors describe the nursing profile? Were these outpatient nurses within the otolaryngology department, nurses focused on head and neck cancer patients?
Response: We thank the reviewer for allowing us to further explain this issue. The dedicated nurses were from the otolaryngology department responsible for the care of head and neck cancer patients. They underwent 1-month specific training for HBET, which was supervised by specialists from the department of surgical oncology and rehabilitation. The dedicated nurses were also well educated so they can properly consider specific issues regarding HBET and ensure the safety of the exercise. After training, these dedicated nurses visited the intervention group in the ward to provide guidance and demonstrate each training movement in HBET before and 2 weeks after the surgery.
In response to the reviewer’s comment, we have added the information regarding the nursing profile in the method section (lines 138-144).
Comment 4: Is there a reason error bars were not included in Figures 2 and 3?
Response: We thank the reviewer for allowing us to further explain this issue. The reason for not including error bars is because the main purpose of these figures is to show the trends of changes in outcome measures over time in the 2 groups (lines 211-215). The presentation of these trends of changes over time was followed by the generalized estimating equation (GEE) analyses of these outcome measures in relation to group and time. Certainly, if we used two-sample independent tests to compare the differences between the 2 groups at any given follow-up timepoint, the error bars would be required. Also, since observing the trends of changes in outcome measures over time in the 2 groups is the focus of presenting these two figures, we made the decision to simplify the figures and did not wish this focus masked by the error bars.
In response to the reviewer’s suggestion, we have redrawn figures 2 and 3 for better quality and reported values of mean ± SD in the new Figures 2 and 3. We sincerely hope that the reviewer could approve our explanation and correction.
Comment 5: Was ECOG/Zubrod score similar between the 2 groups? ECOG/Zubrod is usually noted in quality of life studies.
Response: The reviewer has asked a very interesting question. Indeed, the ECOG/Zubrod score is often used to assess the performance status in patients with cancer, particularly regarding their actual level of function and ability of self-care. Unfortunately, we did not use this assessment in this study and are unable to know the group difference in the ECOG/Zubrod score. Instead, we used the University of Washington Quality of Life Questionnaire (UW-QOL), one of the most commonly used questionnaires in patients with head and neck cancers (reference 36 in the manuscript).

Round 2
Reviewer 3 Report
The authors have revised their paper appropriately.